# Shear force effect of the dry process on cathode contact coverage in all-solid-state batteries

Dongkyu Lee[1], Yejin Shim[1,2], Youngsung Kim[3], Guhan Kwon[3], Seung Ho Choi [2] ✉, KyungSu Kim [2] ✉ & Dong-Joo Yoo [1] ✉

The state-of-the-art all-solid-state batteries have emerged as an alternative to the traditional flammable lithium-ion batteries, offering higher energy density and safety. Nevertheless, insufficient intimate contact at electrode-electrolyte surface limits their stability and electrochemical performance, hindering the commercialization of all-solid-state batteries. Herein, we conduct a systematic investigation into the effects of shear force in the dry electrode process by comparing binder-free hand-mixed pellets, wet-processed electrodes, and dry-processed electrodes. Through digitally processed images, we quantify a critical factor, 'coverage', the percentage of electrolyte-covered surface area of the active materials. The coverage of dry electrodes was significantly higher (67.2%) than those of pellets (30.6%) and wet electrodes (33.3%), enabling superior rate capability and cyclability. A physics-based electrochemical model highlights the effects of solid diffusion by elucidating the impact of coverage on active material utilization under various current densities. These results underscore the pivotal role of the electrode fabrication process, with the focus on the critical factor of coverage.

Among the most remarkable advancements of recent technology is the proliferation of portable electronic devices and the adoption of electric vehicles (EVs) to pave the way toward a greener and more sustainable future instead of the use of fossil fuels[1]. However, with this exponential growth in energy demand, concerns about battery safety have emerged as a critical challenge that demands immediate attention. Conventional lithium-ion batteries (LIBs) with flammable liquid electrolytes, though efficient, have shown vulnerability to thermal runaway events, posing safety risks that call for a paradigm shift in energy storage technologies[2,3]. In this regard, all-solid-state batteries (ASSBs) have emerged as a groundbreaking solution with the potential to change the landscape of energy storage by delivering superior safety characteristics supported by the advancements of sulfide-based solid electrolytes with high ionic conductivities ($10^{-3}$–$10^{-2}$ S cm$^{-1}$) at room temperature such as $Li_{10}GeP_2S_{12}$[4,5], $Li_6PS_5Cl$[6,7], and $Li_2S$-$P_2S_5$[8,9].

One of the key challenges in ASSBs that hinder their widespread commercialization is the interfacial problem at the electrode–electrolyte interface. The interface plays a crucial role in the efficient transport of lithium ions between the solid electrolytes and the active materials, directly influencing the battery's overall performance and stability. At an atomic scale, the (electro) chemical instability between the solid electrolytes and the high-nickel cathode active materials forms resistive byproducts at the interface, leading to increased impedance, reduced ion mobility, and ultimately hampered rate capability[10–14]. The chemical reactivity also triggers structural degradation of active materials, resulting in deteriorated capacity retention and long-term reliability[10,15]. A limited contact between the active material and the solid electrolyte further exacerbates these problems. While numerous studies have been conducted on coating materials with high chemical stability, such as $LiNbO_3$[16–18], $Li_2ZrO_3$[18,19],

[1]School of Mechanical Engineering, Korea University, Seoul, Republic of Korea. [2]Advanced Batteries Research Center, Korea Electronics Technology Institute, Seongnam, Republic of Korea. [3]Production Engineering Research Institute, LG Electronics Incorporation, Seoul, Republic of Korea. ✉ e-mail: sh.choi@keti.re.kr; kimkyungsu@keti.re.kr; djyoo@korea.ac.kr

LiTaO$_3$[20,21], and Li$_3$PO$_4$[22], the coating procedure is not only time- and cost-intensive but also poses an increased interface impedance due to the low ionic conductivity of coating materials.

An equally critical challenge that significantly limits the performance of ASSBs is the solid (point) contact problem between the solid electrolytes and the active materials. This issue arises from poor interfacial adhesion and limited intimate contact between the two components, hindering efficient charge transfer and ion diffusion[13,23–26]. The formation of non-conductive interfaces, voids, or delamination zones restricts the effective diffusion of lithium ions, leading to higher interfacial resistance and limited utilization of active materials in the electrodes. In addition, the solid contact problem promotes localized stress concentration and mechanical strain during charge-discharge cycles due to the volume expansion and contraction of layered cathodes (-5.7%)[27,28], contributing to premature electrode degradation and capacity fading. To overcome this obstacle and utilize the full capacity of ASSBs, novel techniques for electrode fabrication were explored. These approaches encompassed the utilization of ionic liquid-applied electrodes[29], (solution- or suspension-based) wet processes[30–32], and dry processes[32,33].

The dry electrode process has been actively explored for cathodes in LIBs instead of the suspension-based wet process because it enables high-loading cathodes (>5 mAh cm$^{-2}$) for high energy density[34,35]. The dry process has been recently applied to cathodes for ASSBs. While the solvent-drying step of wet processes causes agglomeration of carbon black powders and cracking in high-loading conditions due to binder migration, the solvent-free dry process permits homogeneous mixing and intimate contact of active materials with solid electrolytes, resulting in improved rate capability and cyclability[36,37]. However, the mechanism behind the high performance of dry-processed electrodes has not been revealed; rather, it has been attributed to the increased intimate contact.

Herein, we systematically investigate the shear force effect of the dry process by comparing hand-mixed pellets without binder, wet-processed electrodes, and dry-processed electrodes. Through the microscopy image process, we separate the regions of active material, solid electrolyte, and void, extracting a new critical factor of coverage, which is the area ratio of active materials contacting solid electrolytes. In the case of dry-processed electrodes, the coverage was significantly higher than that of wet-processed ones due to the deformation of the ductile sulfide solid electrolyte. The coverages of each electrode were matched with rate capability compared to that of full coverage in liquid electrolytes. In addition, with the fact that the most sluggish lithium-ion transport is solid diffusion in active material, it is revealed by a physics-based electrochemical modeling that the coverage plays a critical role in determining rate capability in a way that it limits solid diffusion into the interior, exerting a high overpotential. This paper offers useful insight into the relation between the electrode fabrication process and battery performance, particularly with a focus on the coverage of active materials.

## Results
### Characteristics of wet- and dry-processed electrodes
Figure 1 elucidates the disparity between electrodes fabricated via wet and dry processes. In wet-processed electrodes, as the solvents evaporate, binders are coated onto the active material surfaces while carbon black conductors form aggregates. Conversely, dry-processed electrodes, owing to their absence of solvents along with the application of shear force, result in limited binder coating on the active material surfaces. Notably, shear force exerted during the fabrication process deforms the solid electrolyte particles, inducing enhanced coverage compared to the wet-processed electrodes. This stands in contrast to wet-processed electrodes where point contact is made between active materials and solid electrolytes. Considering that the most sluggish step during charging-discharging is solid diffusion in

active materials[38,39], the enhanced coverage of dry-processed electrodes could facilitate lithium diffusion and high utilization (capacity) of active materials at high current densities.

### Extraction and quantification of coverage
We started by fabricating three types of electrodes: a hand-mixed pellet, a wet-processed electrode, and a dry-processed electrode, as depicted in Supplementary Fig. S1. To observe how the different fabrication processes affect the electrodes, cross-sectional scanning electron microscope (SEM) images were acquired for each electrode. The severe aggregation of carbon black conductors in the pellet electrode was observed, while the wet-processed electrode showed an improved dispersion in the energy-dispersive X-ray spectroscopy (EDS) image (Supplementary Figs. S2 and S3). In contrast, the EDS image of the dry-processed electrode (Supplementary Fig. S4) reveals an impressive degree of dispersion among the components due to the absence of the solvent-evaporating step which forces carbon black conductors aggregate again.

Furthermore, it is also worth mentioning that the active materials of the dry-processed electrode exhibit an improved level of intimate contact with the solid electrolytes compared to the pellet and the wet-processed electrode counterparts. This distinction is more evident in Supplementary Figs. S5 and S6, where we compared wet-processed and dry-processed electrodes before applying external pressure. There appear to be a lot of void regions surrounding the active materials and aggregates of the solid electrolyte particles within the wet-processed electrode (Supplementary Fig. S5). On the other hand, the active materials of the dry-processed electrode show a significantly improved level of contact with the solid electrolyte particles (Supplementary Fig. S6). This indicates that the shear force effect plays a critical role in particle-to-particle contact by deforming the ductile sulfide solid electrolyte particles.

To quantify coverage depending on the fabrication process, the contrast and the brightness of the cross-sectional SEM images were adjusted to ease the proceeding steps (Supplementary Fig. S7). The images were then digitally processed to distinguish active material, void, and solid electrolyte regions. Regions characterized by lighter shades of gray were designated in yellow, indicating the circumference of the active materials, while black regions were colored red to separate voids from the solid electrolyte regions (Fig. 2 and Supplementary Fig. S8). The circumference of the electrolyte-covered regions and void-covered regions were measured to calculate the average coverage of each electrode. Specifically, the average coverage of the pellet, wet-processed electrode, and dry-processed electrode were 30.6%, 33.3%, and 67.2%, respectively. The dry-processed electrode clearly demonstrates significantly greater coverage in comparison to all other electrode configurations.

### Resistance analyses of electrode configurations
In order to systematically investigate the ramifications of coverage on the electrochemical performances of cells employing different electrode configurations, we fabricated Li-In∥LiNi$_{0.5}$Mn$_{0.3}$Co$_{0.2}$O$_2$ (NMC532) pressure cells. The same type of Li$_6$PS$_5$Cl (LPSCl) solid electrolytes used in the wet- and dry-processed electrodes were pressed into pellets to be used as the bulk electrolyte. The area of the pellet, wet-processed, and dry-processed electrodes were 1.327 cm$^2$, 1.317 cm$^2$, and 1.317 cm$^2$, respectively. The areal mass loadings of the cathode active materials were 23.5 mg cm$^{-2}$. Further information regarding cell fabrications can be found in the Methods section. We first started by measuring the internal resistance of these cells using electrochemical impedance spectroscopy (EIS), as shown in the Nyquist plot of Fig. 3a. The wet-processed electrodes show noticeably large semi-circles, or internal resistances compared to their counterparts at both 25 °C and 60 °C. The first semi-circles for pellets and dry-processed electrodes can be thought as the resistance in the

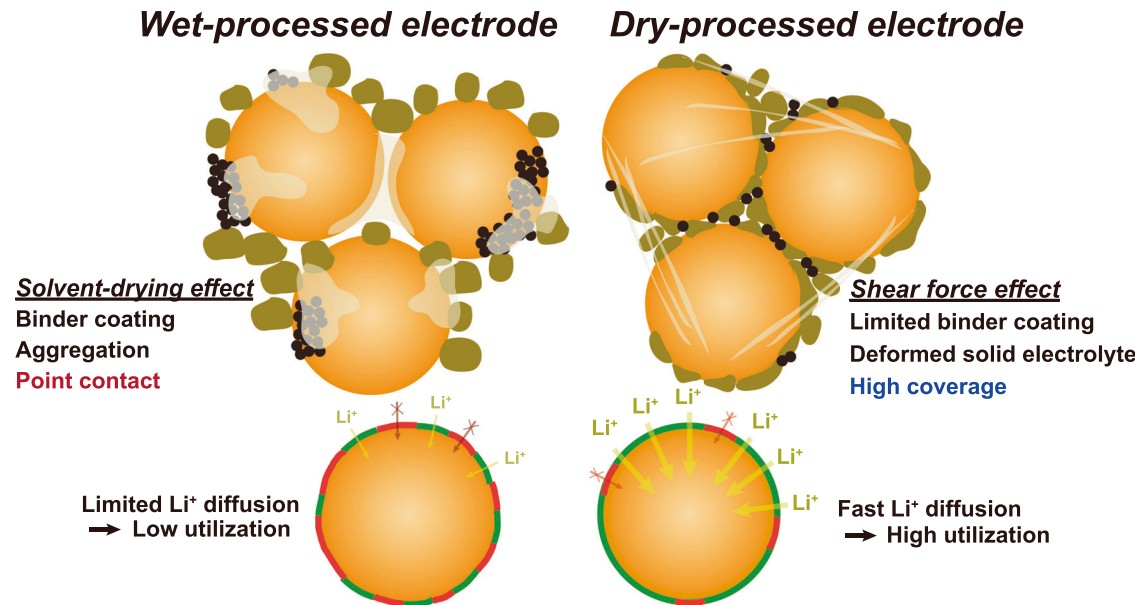

**Fig. 1 | Disparity between wet- and dry-processed electrodes.** Schematic illustration of the difference between wet-processed electrodes and dry-processed electrodes.

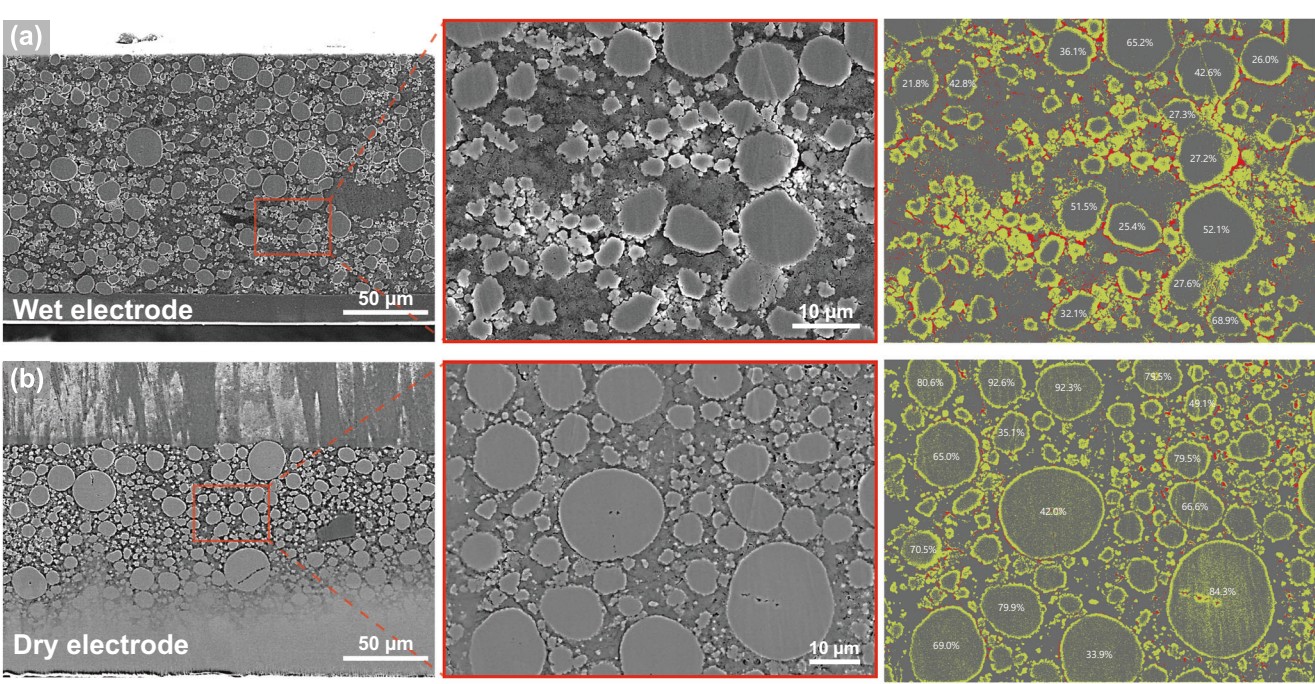

**Fig. 2 | Extraction and quantification of coverage.** Cross-sectional scanning electron microscope (SEM) images of **a** wet- and **b** dry-processed electrodes in different scale bars of 50 μm and 10 μm. The digitally processed images for coverage measurement of each electrode.

anode–electrolyte interface, while the following semi-circles can be seen as the cathode–electrolyte interface resistance[40]. For the wet electrode, the similar time constant for both cathode and the anode interface results in a mixed semicircle[40]. A similar trend for internal resistances can be seen when tested at an elevated temperature of 60 °C. The high temperature accelerates ion and charge transport, therefore decreasing the overall resistance in all configurations. The distribution of relaxation time (DRT) method was conducted to deconvolute the various impedances mixed up in the EIS data, as shown in Supplementary Fig. S9. The DRT data can be divided into four different frequency regions[41,42]: Warburg diffusion ($10^{-3}$–$10^{-1}$ Hz), solid electrolyte–anode interface ($10^{-1}$–$10^{1}$ Hz), solid electrolyte–cathode interface ($10^{1}$–$10^{4}$ Hz), and solid electrolyte grain boundaries

($10^{4}$–$10^{6}$ Hz). Similar to the EIS data, the DRT analysis also points to the large electrolyte–cathode interface peak in wet-processed electrodes (Supplementary Fig. S9b, e), indicating that the large semicircle in wet electrodes is attributed to the electrolyte–cathode interface.

To examine if the chemical stability between solvents and solid electrolytes were the cause of such differences between electrode configurations, experiments to test for possible chemical degradations were conducted. As shown in Supplementary Fig. S10, no changes in the color of electrolyte dispersed solution were found after being rested for 24 h, indicating minimal chemical reactions. To further investigate for signs of degradation, the solution was dried at 80 °C under vacuum, in an effort to retrieve the solvent-exposed electrolyte, to be examined in comparison with pristine solid electrolytes.

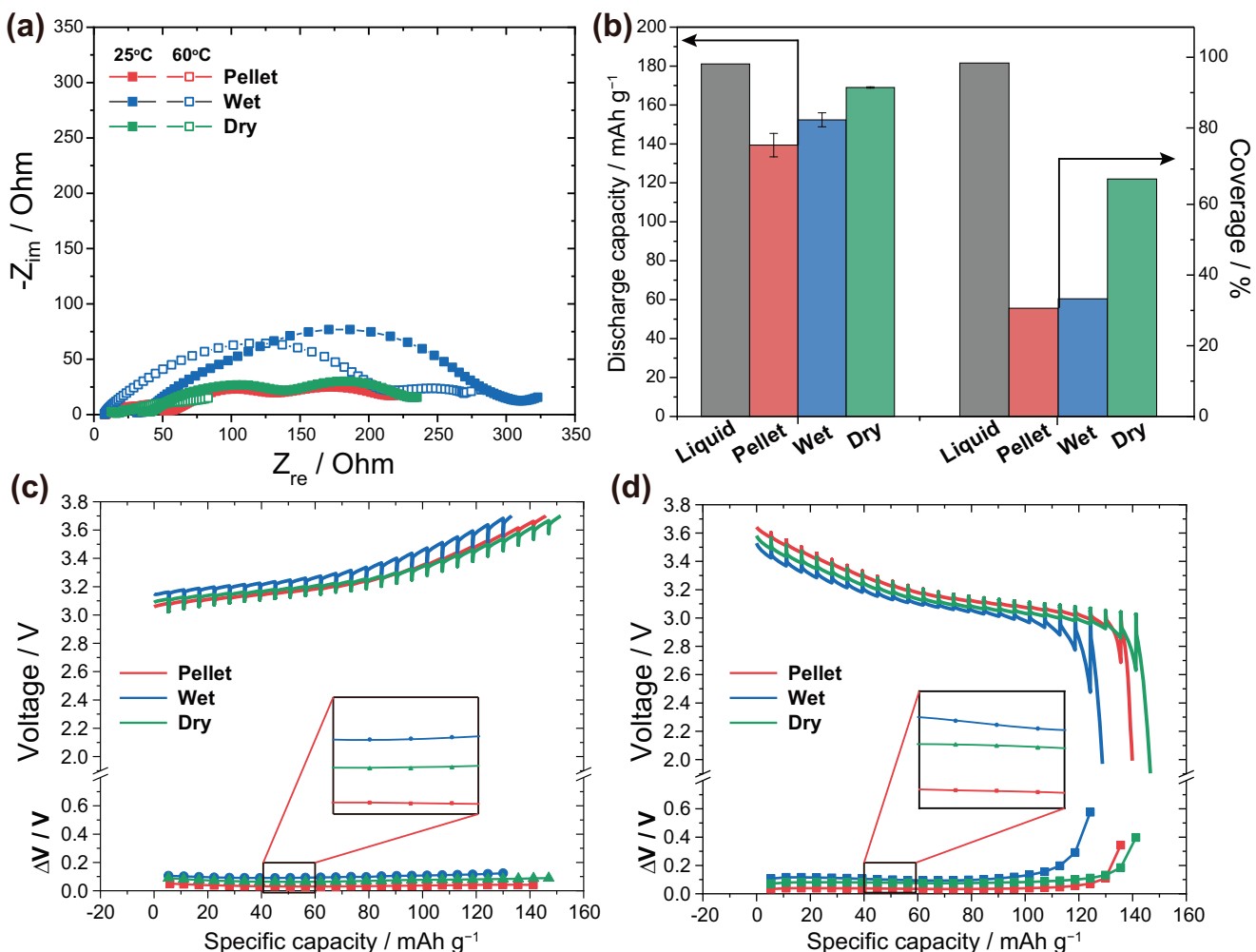

**Fig. 3 | Resistance analyses of electrode configurations. a** Nyquist plots of pellet, wet, and dry electrodes at 25 °C and 60 °C. **b** Discharge capacities at 0.1 C (17 mA g$^{-1}$) and coverages of each electrode with different process. The error bars indicate standard error. Galvanostatic intermittent titration technique (GITT) curves of pellet, wet, and dry electrodes during (**c**) charging or **d** discharging with voltage polarization (ΔV) at 0.2 C (34 mA g$^{-1}$). Li-In∥LiNi$_{0.5}$Mn$_{0.3}$Co$_{0.2}$O$_2$ pressure cells. The mass loading and areal capacity of the cathode were 23.5 mg cm$^{-2}$ and 4.23 mAh cm$^{-2}$, respectively. All the cells were tested at 60 °C.

Both X-ray diffraction (XRD) data and X-ray photoelectron spectroscopy (XPS) reveal no signs of chemical degradations of the solid electrolytes (Supplementary Fig. S11). Furthermore, we fabricated pressure cells to obtain the ionic conductivity of pelletized bulk electrolytes at 25 °C. The results (Supplementary Fig. S12) show negligible difference in ionic conductivity between pristine and solvent-exposed solid electrolytes. It can be drawn that the solvent was not responsible for the high impedance at the cathode interfaces of the wet-processed electrode. Hence, the difference in impedances can be attributed to the coating of binders onto the active materials for wet electrodes. Naturally, when comparing the internal resistance of pellets and dry-processed electrodes, the variance in coverage did not yield a proportional impact on the internal resistance. Despite the large difference in coverage, the internal resistance of pellets and dry-processed electrodes remained relatively similar, with the pellet being slightly smaller. This can be attributed to the fact that while the non-covered regions limit electrochemical reaction sites, the absence of binders relatively facilitates the electrochemical reactions in the covered regions.

Although there is a collective impact of coverage and binders on the electrodes, an evident correlation between coverage and discharge capacity is depicted in Fig. 3b. These cells, after being fully charged, were discharged at a low current density (0.1 C, 17 mA g$^{-1}$). The cells were tested at an elevated temperature of 60 °C to mitigate the negative effects of the electrolyte's low ionic conductivity at low temperatures, and to utilize the full potential of the electrodes. Furthermore, to compare the cells to a more ideal cell with a coverage of 100%, a coin cell was fabricated using liquid electrolytes (1 M LiPF$_6$ in EC/DEC = 1/1). While the liquid electrolyte coin cell shows discharge capacity of 181.1 mAh g$^{-1}$ at 100% coverage, pressure cells made with dry-processed electrodes also showed high discharge capacity, averaging at 169.0 mAh g$^{-1}$ (sample size = 2, standard deviation = 0.453 mAh g$^{-1}$) with 67.2% coverage. Cells with wet-processed electrodes and pellets showed lower discharge capacities of an average of 152.4 mAh g$^{-1}$ (sample size = 4, standard deviation = 7.2 mAh g$^{-1}$) and 139.4 mAh g$^{-1}$ (sample size = 7, standard deviation = 15.95 mAh g$^{-1}$), respectively, at 33.3% and 30.6%, correspondingly. The correlation between coverage and discharge capacity elucidates the pivotal role of coverage in the electrochemical performances of ASSBs.

Upon analyzing the impedances and discharge capacities of pellet and wet-processed electrodes as depicted in Fig. 3a, b, the trend can seem off-putting. Normally, at low C-rates where the performance of cells approaches that of the thermodynamic state, the levels of impedances correspond to the capacities, which contradicts the observed results. This can be thought to be the effect of coverage and the

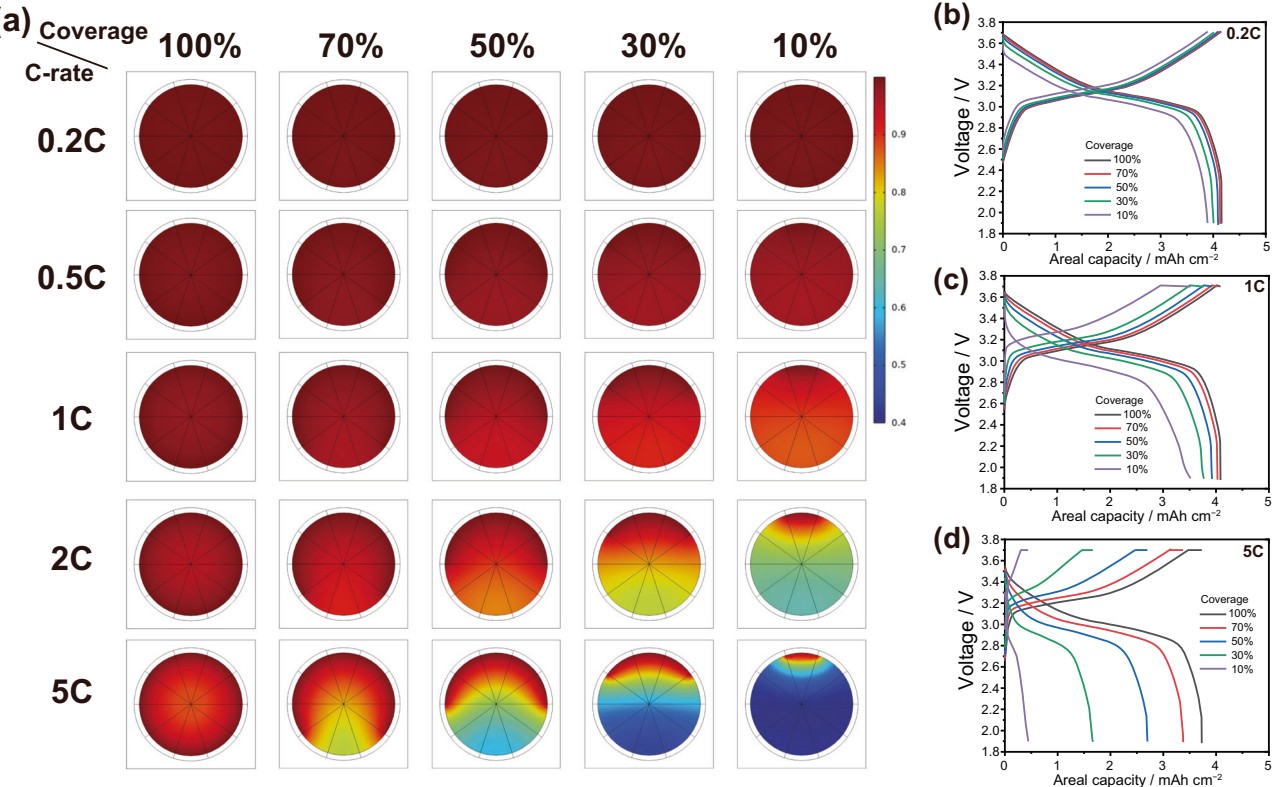

**Fig. 4 | Electrochemical modeling of coverage-induced behaviors.** Electrochemical modeling results of single lithium nickel manganese cobalt oxide (NMC) particle with different coverages and C-rates. **a** Lithium-ion concentration contours in the NMC particles at the end of discharging. Voltage profiles at different C-rates of **b** 0.2 C (34 mA g⁻¹), **c** 1 C (170 mA g⁻¹), and **d** 5 C (850 mA g⁻¹). The modeling parameters: R = 10 μm, cond. = 1 mS cm⁻¹, and clumped configuration, where R indicates the radius of the particle and cond. indicates the ionic conductivity of solid electrolytes.

distribution of coverage, as shown in Supplementary Fig. S13. Although the pellet and wet-processed electrodes exhibit similar levels of coverage, the active material particles of the pellet electrodes show more particles with less than 20% coverage. We believe that particles under such extreme conditions are incapable of being fully utilized, hence, the lower discharge capacities of pellet electrodes despite the lower impedance compared to the wet-processed electrodes.

We believe that due to the restricted reaction sites caused by poor coverage, diffusion within the active material particle plays a crucial role in limiting capacity utilization. To illustrate this, we conducted galvanostatic intermittent titration techniques (GITT) analysis on a fresh set of pressure cells. In Fig. 3c, d, the voltage profile and voltage polarization of the charge and discharge steps in GITT is plotted, respectively. The voltage polarization is the difference between the transient voltage and the steady-state voltage. For both the charge and discharge, the dry electrode shows a lower polarization compared to the wet-processed electrode. This is likely ascribed to the effect of binder coating and the difference in coverage. On the other hand, while the pellet electrode exhibited the lowest value of voltage polarization due to the absence in binders, it showed a limited capacity utilization with its low coverage. In addition, the dry-processed electrode shows the largest discharge capacity.

## Electrochemical modeling of coverage

In pursuit of enhanced understanding of solid-state diffusion taking place within active material particles, we developed a physics-based electrochemical model. Inspired by the one-dimensional single-particle model (SPM)[43,44], we simplified the representation of cathode active material particles to a singular particle. To visualize the activity of lithium ions within the particle, we expanded the space dimension to two dimensions. To mimic the effects of voids and coverage, a thin

layer of electrolyte surrounding the particle was introduced as shown in Supplementary Fig. S14, where the uncovered regions are highlighted in red. Covered regions were modeled the same way as the electrolyte domains, while voids were modeled to insulate any electron or ion transport. The thickness of the cathode is decreased, allowing the particle to barely fit into the bulk electrode domain. Such simplifications, although they hamper the ability of the model to precisely pursue real-life cell performance, highlight the effects of coverage while eliminating other effects such as ionic conduction in the bulk electrode domain. In an effort to reproduce the geometry of real-world cells, two candidates for the coverage layer were considered: a clumped arrangement and a spaced arrangement of voids. Upon close examination of the SEM images (Fig. 2 and Supplementary Fig. S8), we discerned that voids were clumped together, rather than being uniformly distributed across the active material surfaces. Consequently, the clumped version was deemed to be a more accurate representation. Despite the fact that this model cannot precisely match the experimental results, it provides valuable insight into the impact of coverage in ASSBs. A brief comparison of the two models will be conducted in the subsequent sections.

Figure 4a shows how coverage affects active material utilization under different C-rates. Cells featuring diverse coverage levels of 10%, 30%, 50%, 70%, and 100% were charged and discharged at various C-rates (specific current densities) of 0.2 C (34 mA g⁻¹), 0.5 C (85 mA g⁻¹), 1 C (170 mA g⁻¹), 2 C (340 mA g⁻¹), and 5 C (850 mA g⁻¹). The contour represents the degree of lithium concentration, or, the state-of-lithiation (SOL) within the active material particle at the end of discharge. Regions denoted by dark red indicate well-discharged zones, while dark blue indicates regions with where diffusion was relatively incomplete. It is evident that particles featuring different coverages show disparate behaviors. At lower C-rates of 0.2 C

(34 mA g$^{-1}$) and 0.5 C (85 mA g$^{-1}$), particles across all coverage levels exhibit decent uniformity in SOL, with only a slight decrease in discharge capacities for particles with lower coverage levels (Fig. 4b). This is likely because the solid diffusion inside the active material is fast enough to compensate for the limited reaction sites at the low current densities. However, with the increase of C-rate to 1 C (170 mA g$^{-1}$), a subtle gradient in concentration becomes apparent in particles featuring 10% and 30% coverage, indicating suboptimal utilization of active materials. The non-uniformity in concentration directly influences the discharge capacity, as shown in Fig. 4c. For particles characterized by 10% and 30% coverage, their respective discharge capacities at 1 C are 3.52 mAh cm$^{-2}$ and 3.77 mAh cm$^{-2}$, with capacity retentions of 86%, and 92%, correspondingly. We believe that this is the tipping point where solid-state diffusion starts to give in, limiting the full-utilization of active materials. The non-uniformity in ion concentration becomes even more pronounced as the C-rate is increased to 2 C (340 mA g$^{-1}$) and 5 C (850 mA g$^{-1}$) (Fig. 4d). At 2 C (340 mA g$^{-1}$) charge and discharge, the particles featuring 10% and 30% coverage show noticeable ion concentration gradient. Similarly, particles with 50% and 70% coverage also display slight gradients. Subsequently, upon further escalation of C-rate to 5 C (850 mA g$^{-1}$), the 10% and 30% covered particles exhibit a marginal utilization. Likewise, particles with 50% and 70% coverage also exhibit limited utilization of active materials. At the C-rate of 5 C (850 mA g$^{-1}$), the discharge capacities of particles with 10%, 30%, 50%, 70%, and 100% are 0.44, 1.67, 2.70, 3.38, and 3.73 mAh cm$^{-2}$, with capacity retention of 12%, 45%, 72%, and 91%, respectively. Hence, the difference in lithium diffusion and utilization of active material at high current densities is due to the difference in coverage. Electrodes with low coverage (pellet and wet) have restricted reaction sites and are forced to utilize the non-reacting parts of the active material by diffusion. In extreme cases like 10% coverage, the lithium must diffuse as much as the diameter of the active material to be fully utilized. Dry-processed electrodes with high coverage, on the other hand, have a much shorter diffusion range. When current densities are low the diffusion is not as limiting, while at high current densities, the slow diffusion becomes the limiting factor, therefore lowering the discharge and charge capacities.

Furthermore, a parameter study was performed for two parameters: the ion conductivity of the bulk electrolyte and the radius of the active material particle, as depicted in Supplementary Figs. S15–S18. The difference in clumped and spaced voids were also tested (Supplementary Fig. S19). The ionic conductivity parameters were 0.2 mS cm$^{-1}$, 1 mS cm$^{-1}$, and 5 mS cm$^{-1}$, while the radius parameters were 2 µm, 5 µm, and 10 µm. The change in ion conductivity showed no difference in the concentration gradients (Supplementary Figs. S15 and S16). This indicates that solid diffusion is the limiting factor for the utilization of active materials. When the particle radius was changed, the concentration gradient was noticeably relaxed as the radius declined (Supplementary Figs. S17 and S18). Specially, when the radius was decreased to 2 µm, even the particle with 10% coverage showed complete utilization of active materials. Also, the particle with evenly spaced coverage showed enhanced utilization (Supplementary Fig. S19). These findings emphasize the importance of diffusion lengths within the particle.

Since the single-particle model eliminates the effects of ion conduction in the bulk electrode, the need for expansion in the thickness direction of the electrode seemed necessary. Hence, an expanded multi-particle model (MPM) was developed to account for the heterogeneity within the electrode and active material particles, assuming uniform periodic packing and size distribution of particles. Although the assumptions made limit detailed modeling of the electrode geometry, the model suffices to account for the effects of ion conduction. Supplementary Fig. S20a illustrates the concept of the MPM, elucidating the expansion in both the lateral and longitudinal directions. Considering the fact that the direction of movement of the Li ions

during battery operation is primarily in the lateral direction, the expansion in the longitudinal direction is unnecessary. The simulation results show a negligible difference between a single-layer, four-layer, and eight-layer model, as depicted in Supplementary Fig. S20b. The following simulations will therefore be conducted with the single-layer MPM model.

Upon careful analysis of the results of the MPM (Supplementary Fig. S21) in comparison to the original SPM, the gradient of the state-of-lithiation in the lateral direction is pronounced. This can be thought as the effects of ionic conduction in the electrode, which was eliminated in the SPM. The particles on the left (close to the current collector) are shown to be less utilized compared to the particles closer to the bulk electrode. Such a trend is more noticeable at higher coverages, where particles with lower coverage show less of a trend due to the generally low utilization. However, the particle closest to the bulk electrolyte (the rightmost particle) closely resembles the particle from the SPM model. This is attributed to the fact that these particles are richer in Li-ion supply as compared to the particles located deeper into the electrode (the leftmost particles). Although very briefly, this model considers the ionic conduction in the thickness direction of the electrode and still displays the effects of coverage and solid-state diffusion within the active material particles. In addition to the model expansion, the orientation of the particle coverage was considered, as depicted in Supplementary Fig. S22. Each particle was given a random degree of rotation to analyze whether the direction of coverage affected the degree of active material utilization. The magnitude of the state-of-lithiation was identical to the non-rotated model, affirming minimal effects of coverage orientation. This is likely due to the particles being too small to induce a meaningful lateral gradient.

### Electrochemical performances of electrode configurations

The electrochemical performance of the cells fabricated using each type of electrodes were evaluated at 25 °C and 60 °C (Fig. 5a, b). As predicted by the simulation results, the dry-processed electrodes show an exceptional capacity and rate capability, owing to its high-coverage level of 67.2% induced by shear force effect. At a low rate of 0.1 C (17 mA g$^{-1}$), the cells exhibited capacities 150 mAh g$^{-1}$ and 169 mAh g$^{-1}$ at 25 °C and 60 °C, respectively. The wet-processed electrode exhibits discharge capacities of 136 mAh g$^{-1}$ and 152 mAh g$^{-1}$ at 25 °C and 60 °C, respectively. The pellet electrode shows capacities of 126 mAh g$^{-1}$ and 139 mAh g$^{-1}$ which was the lowest amongst all types of electrodes. However, the tendency of discharge capacities is slightly reversed at higher C-rates for the wet-processed and the pellet electrodes, where the former shows capacity retention of 13% and 31%, and the latter 53% and 78% at 25 °C and 60 °C at 1 C (170 mA g$^{-1}$), respectively. Coin cells with liquid electrolytes were also tested, exhibiting high capacity and decent rate capability, as depicted in Supplementary Fig. S23. However, due to the severe degradation induced by the Li metal anodes and the exposure of the liquid electrolyte to high temperature, no further analyses were conducted.

The steep decline in specific capacities of dry electrodes at 60 °C after 1 C (170 mA g$^{-1}$) is very noticeable, compared to the pellet electrodes (Fig. 5b). At a lower temperature of 25 °C, the rate-determining factor for both configurations were thought to be solid-state diffusion within the particle. However, as the temperature was elevated, both diffusion and charge transfer rate are increased. We have concluded that the distinction in rate capability at different temperatures is due to the two factors competing, as the binder-free pellet electrode exerts more capacity at higher C-rates at 60 °C. While the increased performances of dry-processed electrodes are due to their high coverage, the difference in the wet-processed and pellet electrodes are thought to be due to binders. In Supplementary Fig. S24a, the voltage profiles at 0.1 C (17 mA g$^{-1}$) and at 25 °C is depicted. A large overpotential at the initiation of the discharge (charge) cycle is shown for wet electrodes. Conversely, the degree of the overpotentials for the pellet and dry

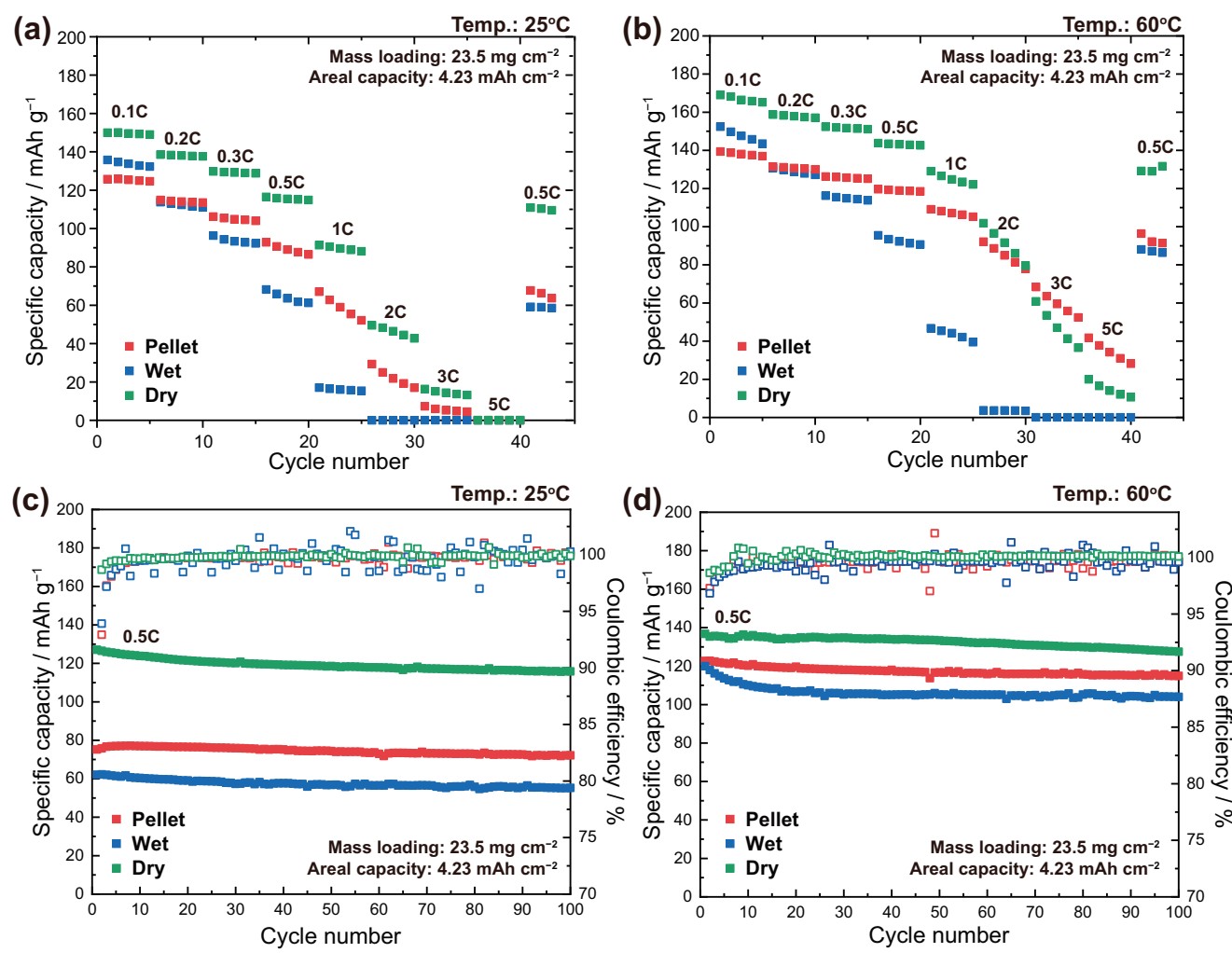

**Fig. 5 | Electrochemical performances of electrode configurations.** Rate capability of pellet, wet, and dry electrodes at **a** 25 °C and **b** 60 °C. Cycling test results at **c** 25 °C and **d** 60 °C. Li-In||LiNi$_{0.5}$Mn$_{0.3}$Co$_{0.2}$O$_2$ pressure cells. The mass loading and areal capacity of the cathode were 23.5 mg cm$^{-2}$ and 4.23 mAh cm$^{-2}$, respectively.

electrodes were similar. The results correspond to the charge transfer resistance analyzed by the EIS. This phenomenon is somewhat relieved when cycled at 60 °C (Supplementary Fig. S24b), while still showing a similar trend between the electrode configurations.

When the cells are tested at an elevated temperature of 60 °C, the overall overpotential is relieved due to increased charge transfer, mass transfer, and diffusion rates. In Supplementary Fig. S25, voltage profiles at various C-rates are depicted. A noticeable increase in discharge capacity and rate capability at elevated temperatures is shown for all electrode types. To systematically study the difference in overpotential for each configuration and C-rate, we quantified the degree of overpotential by the difference between charging voltage and discharging voltage at 50% state-of-charge (SOC) (Supplementary Fig. S26). Both Supplementary Fig. S26a, S26b elucidates the large overpotential in wet electrodes, likely due to the low coverage and the high internal resistance. At 25 °C, the trend for overpotential of pellet and dry electrodes changed with the C-rates. At lower C-rates, the pellets exhibited higher overpotential, while the dry electrodes exhibited higher overpotential at higher C-rates. However, the dry electrodes exhibited greater overpotential compared to the pellets at an elevated temperature of 60 °C. Such phenomenon indicates the competing relationship between charge transfer resistance and coverage. When the cyclability of each cell was evaluated at 25 °C and 60 °C, both at 0.5 C (85 mA g$^{-1}$), the different electrode configurations showed similar levels of capacity retention after 100 cycles, with the

dry-processed electrodes still exhibiting significantly high discharge capacity, especially at 25 °C. (Fig. 5c, d).

## Discussion

Naturally, the liquid electrolyte can be thought to completely cover the active materials. Solid electrolytes on the other hand make point contact with active materials and, owing to the high ductility of sulfide electrolytes, deform to cover the active materials. Without additional processes like mechanical coating[45], the level of coverage is limited. However, according to the modeling results, the limitations due to solid-state diffusion are minimal, with 70% coverage, which is observed in the dry-processed electrodes. In addition, when compared to liquid electrolytes, which have low transference numbers (<0.4), a large amount of the anions move in the opposite direction of the cations and accumulate at the surface of the electrodes, impeding the movement of Li ions. Such movements in ions are more critical at higher C-rates where limitations due to mass transfer become prominent[46]. Conversely, the near-unit transference number of sulfide electrolytes, coupled with the high ionic conductivity[6,7], can overcome the performance of liquid electrolytes when certain coverage conditions are met.

In summary, the shear force effect of the dry process was systematically investigated by comparing hand-mixed pellets, wet-processed electrodes, and dry-processed electrodes. The SEM images of each electrode configuration were utilized to extract and quantify the pivotal factor of coverage. The wet electrodes have larger

internal resistance compared to its counterparts, primarily attributed to the presence of binders and low coverage, proven by EIS and GITT analysis. The exceptionally high coverage in dry electrodes corresponded to a pronounced discharge capacity and rate capability because the utilization of active materials is determined by solid-state diffusion within the particle. This is rationally proved by a physics-based electrochemical model, visualizing the lack of utilization of active materials with low coverages, where the utilization was exacerbated as the C-rate was increased. The present study gives useful insights into the role of electrode processes in all-solid-state batteries.

## Methods

### Electrode fabrication

All electrodes were fabricated in an Ar-filled glove box with oxygen and moisture below 1 ppm. For pellet electrodes, $LiNi_{0.5}Mn_{0.3}Co_{0.2}O_2$ (NMC532), $Li_6PS_5Cl$ (LPSCl), and Super P were hand-mixed in an agate mortar in an 80:18.5:1.5 wt% ratio (a total of 40 mg). The powders were compressed during the cell assembly process, as explained in the subsequent section. For the preparation of wet-processed electrodes, a slurry containing NMC532, LPSCl, Super P, and Neodymium Nitrile Rubber (NdBR) in an 80:17:1.5:1.5 wt% ratio was prepared in butyl-butyrate (BB). The NdBR binder was dispersed in BB using a magnetic stirrer beforehand. The slurry was cast onto a 20 μm thick aluminum current collector. The electrode was dried in a vacuum chamber at 80 °C overnight. The electrode was punched into a circle with a diameter of 12.95 mm for cell fabrication. In the case of dry-processed electrodes, NMC532, LPSCl, and Super P was hand-mixed in an agate mortar. Once the powders were well mixed, polytetrafluoroethylene (PTFE) was added to the compound and mixed until it took a form of a dough. The wt% ratio of the powders was 80:17:1.5:1.5, respectively. The dough-like compound was heated on a hot plate at 85 °C for 5 minutes on each side before being calendered using an aluminum rolling pin. The flattened compound was folded, calendered, and heated five times. The compound was calendered to the appropriate mass loading to be punched into a circle with a diameter of 12.95 mm for cell fabrication.

### Cell assembly

The cells were fabricated in an Ar-filled glove box with oxygen and moisture below 1 ppm. 150 mg of LPSCl powder was pressed at 1.4 TONs using a polyether ether ketone (PEEK) mold with a diameter of 13 mm. On one side of the LPSCl pellet, a pre-made NMC532 electrode and aluminum foil punched in a diameter of 12.95 mm was placed to be pressed at 7 TONs. In the case of the pellet electrode, the powder mixture was spread and tamped before being pressed. For the fabrication of anodes, a 0.1 mm thick indium metal foil punched in a diameter of 12 mm and a 0.2-mm-thick lithium metal foil punched in a diameter of 6 mm was pressed by hand. The lithium-indium anode is placed on the other side of the LPSCl pellet, with the indium side facing the electrolyte. A 20 μm thick SUS foil with a diameter of 12.95 mm was placed on the lithium side before being closed. The cell was then subjected to constant pressure by means of tightening three screws, where each screw was tightened to a torque of 11 Nm.

### LIB electrode fabrication and coin cell assembly

The LIB cathodes were composed of NMC532, Super P, and poly-vinylidene fluoride (PVDF) in a ratio of 94:3:3 wt%. A slurry was fabricated using N-Methyl-2-pyrrolidone (NMP) solvent which were casted on a 20-μm thick aluminum current collector. The electrodes were then dried in a vacuum oven at 60 °C for 12 h. The electrodes were punched into circles with a diameter of 12.95 mm. Lithium metal foil with a thickness of 200 μm was punched into circles with a diameter of 14 mm. 2032-type coin cells were fabricated with the addition of 100 μL of 1 M $LiPF_6$ in EC/DEC (v/v = 1/1) in an Ar-filled glove box with oxygen and moisture below 1 ppm.

### Calculation of ionic conductivity

To calculate the ionic conductivity of the LPSCl electrolytes, cell set-up was designed. A total of 32 mg of LPSCl powder was poured into a PEEK mold with a diameter of 6 mm. The powders were pressed at 1 TON for 3 min. The cells were then subjected to constant pressure by tightening three screws at a torque of 11 Nm. The thickness of the pelletized electrolyte was measured after the EIS analysis, conducted at 25 °C. The ionic conductivity was calculated by the following equation.

$$\text{(Ionic Conductivity)} = \frac{\text{(Thickness of Pellet)}}{\text{(Resistance} \times \text{Surface Area)}} \tag{1}$$

### Characterization and electrochemical measurements

Cross-sectional scanning microscopy image (SEM) was utilized to obtain images of each electrode configuration. The SEM and the EDS images were obtained using a JSM-IT200SEM(JEOL), while the cross-sectional polishing was performed using IB-19520CCP (JEOL). The polishing was performed at 5 kV for 6 h for pellets and 3 and half hours for dry and wet electrodes.

Before any galvanostatic battery tests, all cells were rested for 12 h before three formation cycles were assessed at 0.1 C (17 mA g$^{-1}$) in the voltage window of 1.9–3.7. To measure the internal resistances of each electrode configuration, electrochemical impedance spectroscopy (EIS) analysis was conducted using an impedance analyzer (MP1, Zive). After the formation cycles, the cells were charged until the voltage reached 2.8 V to align all cells at the same state-of-charge (SOC) before the EIS analysis was performed over the frequency range of 10 mHz–800 kHz and an amplitude of 10 mV. The galvanostatic intermittent titration technique (GITT) was performed with 0.2 C (34 mA g$^{-1}$) pulse of 10 minutes, followed by 50 minutes of rest. The fully charged cells were charged and discharged in the GITT mode, sequentially. For the analysis of rate capability, cells were charged and discharged in constant-current–constant-voltage (CCCV) and constant-current (CC) mode with the same C-rate, respectively. For both rate capability and cycling analysis, a battery cycler (WBCS3000L, WonATech) was used. For stable temperature control, a low-temperature incubator (DS-11B, DASOL SCIENTIFIC) and a drying oven (ThermoStable OF-105, DAIHAN SCIENTIFIC) were used for 25 °C and 60 °C, respectively.

### Numerical modeling

All numerical calculations in this paper were performed using COM-SOL Multiphysics software. The lithium-ion battery interface was used to compute the electrochemical reactions occurring inside the battery cells. The transport of the diluted species interface was coupled with the battery interface to simulate the lithium-ion activities inside the active material particle. Most electrochemical and geometrical parameters were set to match the experiments, while some unknown variables were tuned to fit the experimental results. Parameters are listed in Supplementary Table S1. All the details are included in the Supplementary Note 1.

## Data availability

The data that support the findings of this study are available within the paper and its Supplementary Information file. Any other data are available from the corresponding authors upon request.

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

## Acknowledgements

D.-J.Y. acknowledges financial support from a National Research Foundation of Korea grant funded by the Korea government (MSIT) (RS-2023-00252898) and technical support from the Supercomputing Center/Korea Institute of Science and Technology Information (KSC-2022-CRE-0135) and experimental support from the Next Generation Battery Research Center of LG Energy Solution. K.K. acknowledges financial support from the Institute of Civil Military Technology Cooperation funded by the Defense Acquisition Program Administration and Ministry of Trade, Industry and Energy of Korean government under grant No. 22-CM-FC-20.

## Author contributions

D.-J.Y., S.H.C., and K.K. conceived the project. D.L. and Y.S. designed and conducted the experiments. S.H.C.,K.K., G.K., and Y.K. participated in characterizations and analysis. All authors discussed the results. D.L. and D.-J.Y. wrote the manuscript and received comments from the other co-authors.

## Competing interests

The authors declare no competing interests.
