## [Peer Review File · Nature Communications]

REVIEWER COMMENTS

Reviewer #1 (Remarks to the Author):

In principle, the idea behind the study by Dongkyu Lee et al. is very interesting and the manuscript well written. Unfortunately, the study suffers from several fundamental weaknesses and the conclusions drawn by the authors are not evident. The analysis is not thorough enough and relies only on SEM and electrochemical testing, whereas the chemical stability of the materials involved is not considered at all. Other analysis methods need to be used to investigate the chemical degradation. The electrochemical testing results of wet, dry and pellet indicate that the wet processing may have chemically degraded the cathode mixture, the pellet preparation may be optimized further, which means that the dry shows the best performance of the three here, but it is not better than in other publications that use the pellet method. Furthermore, the SPM simulation is oversimplified and not representative of a real cathode. Clear conclusions cannot be drawn from this data.

Figure 2 digital evaluation: image analysis depends on contrast/brightness settings. Images appear to be quite different. What is the influence of the contrast/brightness settings on image analysis?

The part 'Resistance analyses of electrode configurations' needs to describe the cell composition in more detail: which electrolyte is used, cell area, area loading, etc.

Figure 3. Data is inconsistent: a) wet has the highest impedance; b) wet has the middle capacity; c,d) wet has the lowest capacity. Normally cells with higher impedance have lower capacity, so a) and c,d) data are consistent, but the data shown in b) does not match the trend.

How did the authors check whether the BB solvent degrades the materials during processing? The impedance of wet in Figure 3a indicates chemical degradation of the electrolyte in contact with liquid, which may explain the higher resistance and worse cycling performance of wet.

When the authors attribute coverage in a liquid electrolyte to be 100%, does this mean that solid electrolytes cannot compete as coverage can never be better than in liquid electrolyte?

Figures S9-S13. The single particle model is not accurate regarding the influence of the ionic conductivity because the factor where ionic conduction is limiting is when the thickness of the cathode is increased to many particles thickness. The SPM is too simple and the results are not representative for real cathodes where both ion conduction and solid diffusion play a role.

Figure 5. The authors should add the data for the coin cell with liquid electrolyte.

Figure 5c. Capacity of dry is only 113 mAhg⁻¹, in Figure 5b it is 147 mAhg⁻¹, how come?

Figure 5. There are many examples in literature where the pellet achieves higher specific capacity than shown here, how come?

Table S1 states i_{1C} as current density but the value given is 5.44 mAh, which has the wrong unit.

Table S1. Why is thickness of cathode 25 μm and electrolyte 140 μm ? In a real cell the cathode would be thicker than the electrolyte. This shows that the model is not representative of a real cell, as mentioned above.

Reviewer #2 (Remarks to the Author):

This manuscript reports the effects of shear force in the dry electrode process was systematically investigated by comparing binder-free hand-mixed pellets, wet-processed electrodes, and dry-processed electrodes. Through digitally processed images, the authors quantified a critical factor, 'coverage', the percentage of electrolyte-covered surface area of the active materials. The coverage of dry electrodes was significantly higher (67.2%) than those of pellets (30.6%) and wet electrodes (33.3%), enabling superior rate capability and cyclability. Moreover, a physics-based electrochemical model highlighted the effects of solid diffusion by elucidating the impact of coverage on active material utilization under various C-rates. This paper gains insight into the relation between the electrode fabrication process and battery performance, particularly with a focus on the coverage of active materials. This work is possible to be published in Nature Communications after revisions as follows.

1. The authors should give an explanation why the enhanced coverage of dry-processed electrodes could facilitate lithium diffusion and high utilization (capacity) of active materials at high current densities.
2. Why the cells were tested at an elevated temperature of 60°C, rather than at room temperature?
3. In Figure 3a, to well separate the corresponding resistance, the EIS spectra should be fitted.
4. To understand solid-state diffusion taking place within active material particles, the authors developed a physics-based electrochemical model, and a thin layer of electrolyte surrounding the particle was introduced. How about the accuracy of this model compared to the experimental results?
5. The following related literatures on solid-state batteries are suggested to be cited properly, e.g., *Electrochemical Energy Reviews*, 2023, 6, 17. *Nano Energy*, 2017, 33, 363-386. *ACS Appl. Mater. Interfaces*, 2020, 12, 28345-28350.

Response to Reviewer 1

In principle, the idea behind the study by Dongkyu Lee et al. is very interesting and the manuscript well written. Unfortunately, the study suffers from several fundamental weaknesses and the conclusions drawn by the authors are not evident. The analysis is not thorough enough and relies only on SEM and electrochemical testing, whereas the chemical stability of the materials involved is not considered at all. Other analysis methods need to be used to investigate the chemical degradation. The electrochemical testing results of wet, dry and pellet indicate that the wet processing may have chemically degraded the cathode mixture, the pellet preparation may be optimized further, which means that the dry shows the best performance of the three here, but it is not better than in other publications that use the pellet method. Furthermore, the SPM simulation is oversimplified and not representative of a real cathode. Clear conclusions cannot be drawn from this data.

Response: We first appreciate the reviewer for his/her thorough review of our manuscript. We believe that revisions in response to the reviewer's corrections would improve the quality of the manuscript.

1. Figure 2 digital evaluation: image analysis depends on contrast/brightness settings. Images appear to be quite different. What is the influence of the contrast/brightness settings on image analysis?

Response: Under the assumption that different materials are displayed in different shades (brightness or contrast), we tried to set the brightness/contrast settings so that the active materials, electrolytes, and voids could be distinguished. Additionally, the settings of the dry electrodes were slightly tweaked before digital processing, as the voids were not as visible as the pellets or wet electrodes due to their scarcity. The image below shows the distinction between regions in each electrode. We've edited the part in the manuscript to elucidate the process in a bit more detail.

[page 6]

...on the fabrication process, the contrast and the brightness of the cross-sectional SEM images were adjusted to ease the proceeding steps (Figure S7). The images were then digitally processed...

Original images

Refined images

Figure S7. Original and refined SEM images of (a) pellet, (b) wet-processed, and (c) dry-processed electrodes.

2. The part ‘Resistance analyses of electrode configurations’ needs to describe the cell composition in more detail: which electrolyte is used, cell area, area loading, etc.

Response: We want to thank the reviewer for such a generous and detailed comment. We have added more specific explanations regarding the matter.

[page 7]

... we fabricated Li-In||NMC532 pressure cells. The same type of LPSCl solid electrolytes used in the wet- and dry-processed electrodes were pressed into pellets to be used as the bulk electrolyte. The area of the pellet, wet-processed, and dry-processed electrodes were 1.327 cm^2 , 1.317 cm^2 , and 1.317 cm^2 , respectively. The areal mass loadings of the cathode active materials were 23.5 mg cm^{-2} . Further information regarding cell fabrications can be found in the Methods section. We first started ...

3. Figure 3. Data is inconsistent: a) wet has the highest impedance; b) wet has the middle capacity; c,d) wet has the lowest capacity. Normally cells with higher impedance have lower capacity, so a) and c,d) data are consistent, but the data shown in b) does not match the trend.

Response: We understand that at low C-rates, the charging/discharging processes closely resemble the activities at a thermodynamic state, hence, are attributed to the impedance results shown in the EIS data. However, we believe that even at low C-rates where kinetics have less of an effect on discharge capacity, the kinetics of the ions affect utilization of the active materials owing to the effects of coverage. Having said this, we also believe that the 3% coverage difference between the pellet electrode (30.6%) and the wet electrode (33.3%) was not enough to explain the phenomenon.

In response to the reviewer's comment, we took a closer look into the calculation process of coverage to reveal statistical explanations. We took more sample particles of other sections of the electrodes and calculated the frequency of the coverages. As shown in the distribution graph below (Figure S13), while pellets and wet electrodes both have similar active material particles with high coverage (>60%), pellets appear to have more particles with extremely low level of coverage (<20%). The particles with low coverage are thought to barely exhibit any capacity due to the possible isolation from solid electrolytes, explaining the rather unusual results between impedance and discharge capacity in pellet and wet electrodes.

Also, the tests for Figure 3(b) and 3(c-d) were conducted at different C-rates (0.1C and 0.2C, respectively, and we have added the C-rates in the caption of Figure 3). The trend for discharge capacity is reversed at 0.2C for pellets and wet electrodes, hence the inconsistency in (b) and (c-d). We believe that such trends reinforce our thoughts where the impedance does not fully dictate the discharge capacity, rather, there is another factor (coverage) affecting the performance even at low C-rates.

We have added following Figure and explanation.

[page 9]

... performances of ASSBs.

Upon analyzing the impedances and discharge capacities of pellet and wet-processed electrodes as depicted in Figures 3a and b, the trend can seem off-putting. Normally, at low C-rates where the performance of cells approaches that of the thermodynamic state, the levels of impedances correspond to the capacities, which contradicts the observed results. This can be thought to be the effect of coverage and the distribution of coverage, as shown in Figure S13. Although the pellet and wet-processed electrodes exhibit similar levels of coverage, the active material particles of the pellet electrodes show more particles with less than 20% coverage. We believe that particles under such extreme conditions are incapable of being fully utilized, hence, the lower discharge capacities of pellet electrodes despite the lower impedance compared to the wet-processed electrodes.

Figure S13. Distribution of sample particle coverages in pellet, wet-processed, and dry-processed electrodes.

4. How did the authors check whether the BB solvent degrades the materials during processing? The impedance of wet in Figure 3a indicates chemical degradation of the electrolyte in contact with liquid, which may explain the higher resistance and worse cycling performance of wet.

Response: We understand the reviewers' concerns about possible chemical degradation with solvent in wet-processed electrodes. Four additional experiments were conducted to verify the chemical stability of BB solvent and the solid electrolyte: color change, XPS, XRD, and ionic conductivity.

To test for any noticeable chemical reactions between the solvent and the solid electrolyte, we looked for color changes in the solution. Solid electrolytes were added to BB solvents, were shaken, and were rested for 24 hours, to reveal no significant changes to color, indicating chemical stability (Figure S10).

For XPS and XRD analyses, pristine solid electrolytes and BB-soaked solid electrolytes were compared. For the retrieval of electrolytes, the 24h rested solution was dried at 80°C under vacuum. XPS spectra of P 2p, S 2p, and Cl 2p were analyzed to show no chemical degradation after the exposure of solid electrolytes to BB solvent. The XRD data shows consistent results (Figure S11).

In addition to the experiments above, we wanted to test if there were any impedance-wise drawbacks to the wet electrodes. We designed a pressure-cell-setup to calculate the ionic

conductivity of the bulk electrolyte. The ionic conductivity is calculated by the following equation (Figure S12).

$$\sigma_{SE} = \frac{(Thickness)}{(Resistance \times Area)}$$

In conclusion, there appears to be no signs of chemical degradation between the electrolyte and the solvent.

We have added following Figures and explanation.

[page 8]

To examine if the chemical stability between solvents and solid electrolytes were the cause of such differences between electrode configurations, experiments to test for possible chemical degradations were conducted. As shown in Figure S10, no changes in the color of electrolyte dispersed solution were found after being rested for 24 hours, indicating minimal chemical reactions. To further investigate for signs of degradations, the solution was dried at 80°C under vacuum, in an effort to retrieve the solvent-exposed electrolyte, to be examined in comparison with pristine solid electrolytes. Both XPS and XRD data reveal no signs of chemical degradations of the solid electrolytes (Figure 11). Furthermore, we fabricated pressure-cells to obtain the ionic conductivity of pelletized bulk electrolytes. The results (Figure S12) show negligible difference in ionic conductivity between pristine and solvent-exposed solid electrolytes. It can be drawn that the solvent was not responsible for the high impedance of the wet-processed electrode. Hence, the difference in impedances can be attributed to the coating of binders onto the active materials for wet electrodes. Naturally, when comparing...

[page 21_ Methods section]

Calculation of ionic conductivity: To calculate the ionic conductivity of the LPSCI electrolytes, cell set-up was designed. A total of 32 mg of LPSCI powder was poured into a PEEK mold with a diameter of 6 mm. The powders were pressed at 1 TON for 3 minutes. The cells were then subjected to constant pressure by tightening three screws at a torque of 11 Nm. The thickness of the pelletized electrolyte was measured after the EIS analysis. The ionic conductivity was calculated by the following equation.

$$(Ionic\ Conductivity) = \frac{(Thickness\ of\ Pellet)}{(Resistance \times Surface\ Area)}$$

Figure S10. Optical images of pristine butyl butyrate solvent, after adding solid electrolytes, after shaking solution, and after 24 hours of rest.

Figure S11. (a) XRD spectra and XPS spectra of pristine LPSCI and LPSCI after exposure to butyl butyrate solvent (denoted as LPSCI_BB). (b) P 2p, (c) S 2p, and (d) Cl 2p.

Figure S12. (a) EIS spectra and (b) ionic conductivities of pristine LPSCI and LPSCI after exposure to butyl butyrate solvent.

5. When the authors attribute coverage in a liquid electrolyte to be 100%, does this mean that solid electrolytes cannot compete as coverage can never be better than in liquid electrolyte?

Response: We believe that it would be very challenging to achieve 100% coverage due to the nature of solid-solid particle contact. However, the experimental and simulation results indicate that with a high-enough coverage ($>70\%$), the limitations due to solid-state diffusion are minimal. More importantly, the solid electrolytes have their own strengths in kinetics, such as high ionic conductivity and single-ion conduction mechanism, which we believe, when further engineered, will enable superior performances compared to liquid electrolytes.

We have added following explanation and references.

[page 16]

Naturally, the liquid electrolyte can be thought to completely cover the active materials. Solid electrolytes on the other hand make point-contact with active materials and, owing to the high ductility of sulfide electrolytes, deform to cover the active materials. Without additional processes like mechanical coating,⁴⁵ the level of coverage is limited. However, according to the modeling results, the limitations due to solid-state diffusion are minimal, with 70% coverage, which is observed in the dry-processed electrodes. Additionally, when compared with liquid electrolytes, since they have low transference numbers (< 0.4), a large amount of the anions move in the opposite direction of the cations and accumulate at the surface of the electrodes, impeding the movement of Li ions. Such movements in ions are more critical at higher C-rates where limitations due to mass transfer become dominant.⁴⁶ Conversely, the near-unit transference number of sulfide electrolytes, coupled with the high ionic conductivity,^{6, 7} can overcome the performance of liquid electrolytes when certain coverage conditions are met.

...

45. Kim J, *et al.* High-Performance All-Solid-State Batteries Enabled by Intimate Interfacial Contact Between the Cathode and Sulfide-Based Solid Electrolytes. *Advanced Functional Materials* **33**, 2211355 (2023).

46. Stolz L, Hochstädt S, Röser S, Hansen MR, Winter M, Kasnatscheew J. Single-Ion versus Dual-Ion Conducting Electrolytes: The Relevance of Concentration Polarization in Solid-State Batteries. *ACS Applied Materials & Interfaces* **14**, 11559-11566 (2022).

6. Figures S9-S13. The single particle model is not accurate regarding the influence of the ionic conductivity because the factor where ionic conduction is limiting is when the thickness of the cathode is increased to many particles thickness. The SPM is too simple and the results are not representative for real cathodes where both ion conduction and solid diffusion play a role.

Response: Before we discuss the effects of ionic conduction, we would like to point out that the model was originally designed to not accurately pursue the experiment, but to discuss the diffusion inside the active material particles. However, we also believe that the reviewer's criticism on this matter is valid due to the kinetics of Li ions at high C-rates and the simplicity of the model. To take into account the effects of ionic conduction in the electrode domain along with all the effects of coverage and diffusion, we re-designed the model, expanding the over-simplified single-particle model (SPM) to a more-detailed multi-particle model (MPM).

The model is first considered to be expanded in both spatial dimensions: the longitudinal direction and the lateral direction. The latter can be thought to represent the thickness of the electrode. The figure below shows the primary expansion of the model.

Before we get into analyzing the expanded model, an inspection of the effects in the longitudinal direction was conducted. In the model, there appeared to be no noticeable difference in expansions in the longitudinal direction, as shown in the figure below (Figure S20a). This is rather obvious since the major movements of Li ions are in the lateral direction, from the anode to the cathode, and vice versa. The subsequent simulations will consequently be conducted in a single-layer format.

Upon careful analysis of the results of the multi-particle model in comparison to the original single-particle model, the gradient of the state-of-lithiation in the lateral direction is very noticeable (Figure S20b). This can be thought to be the effects of ionic conduction in the electrode, where the particles closer to the bulk electrolyte are richer in Li-ion supply as compared to the particles located deeper into the electrode. Although very briefly, this model considers the ionic conduction in the thickness direction of the electrode and still elucidates the effects of coverage and the solid-state diffusion within the active material particles.

We have added following Figures and explanation.

Figure S20. (a) Schematics of single-particle and multi-particle models. (b) Modeling results of multi-particle models depending on longitudinal direction.

Figure S21. Electrochemical modeling results of multi-particle models with different coverages, c-rates, and same orientations.

Figure S22. Electrochemical modeling results of multi-particle models with different coverages, c-rates, and random orientations.

[page 13]

Since the single-particle model eliminates the effects of ion conduction in the bulk electrode, the need for expansion in the thickness direction of the electrode seemed necessary. Hence, an expanded multi-particle model (MPM) was developed to account for the heterogeneity in the electrode and active material particles. Figure S20a illustrates the concept of the MPM, elucidating the expansion in both the lateral and longitudinal directions. Considering the fact that the direction of movement of the Li ions during battery operation is primarily in the lateral direction, the expansion in the longitudinal direction is unnecessary. The simulation results show a negligible difference between a single-layer, four-layer, and eight-layer model, as depicted in Figure S20b. The following simulations will therefore be conducted with the single-layer MPM model.

Upon careful analysis of the results of the MPM (Figure S21) in comparison to the original SPM, the gradient of the state-of-lithiation in the lateral direction is pronounced. This can be thought as the effects of ionic conduction in the electrode, which was eliminated in the SPM. The particles on the left (close to the current collector) are shown to be less utilized compared to the particles closer to the bulk electrode. Such a trend is more noticeable at higher coverages, where particles with lower coverage show less of a trend due to the generally low utilization. However, the particle closest to the bulk electrolyte (the rightmost particle) closely resembles the particle from the SPM model. This is attributed to the fact that these particles are richer in Li-ion supply as compared to the particles located deeper into the electrode (the leftmost particles). Although very briefly, this model considers the ionic conduction in the thickness direction of the electrode and still displays the effects of coverage and solid-state diffusion within the active material particles. In addition to the model expansion, the orientation of the particle coverage was considered, as depicted in Figure S22. Each particle was given a random degree of rotation to analyze whether the direction of coverage affected the degree of active material utilization. The magnitude of the state-of-lithiation was identical to the non-rotated model, affirming minimal effects of coverage orientation. This is likely due to the particles being too small to induce a meaningful lateral gradient. ...

7. Figure 5. The authors should add the data for the coin cell with liquid electrolyte.

Response: We thank the reviewer for the comment. The same experiment for liquid electrolytes was conducted additionally to add data to the figure. However, it is worth mentioning that the liquid electrolyte half-cells experience severe degradation after a few cycles due to the Li metal anode and the exposure of the liquid electrolyte to high temperature. Also, the difference in the system hampers the exact comparison between the liquid and solid electrolytes; hence, we've added the data to the supplementary information.

We have added following Figure and explanation.

Figure S23. Rate capability of cells with liquid electrolyte (a) 25°C and (b) 60°C. The mass loading and areal capacity were 23.5 mg cm⁻² and 4.23 mAh cm⁻², respectively.

[page 14]

... 1C, respectively. Coin cells with liquid electrolytes were also tested, exhibiting high capacity and decent rate capability, as depicted in Figure S23. However, due to the severe degradation induced by the Li metal anodes and the exposure of the liquid electrolyte to high temperature, no further analyses were conducted. ...

[page 18_Methods section]

LIB Electrode fabrication and coin cell assembly: The LIB cathodes were composed of NCM523, Super P, and polyvinylidene fluoride (PVDF) in a ratio of 94:3:3 wt%. A slurry was fabricated using N-Methyl-2-pyrrolidone (NMP) solvent which were casted on a 20 μm thick aluminum current collector. The electrodes were then dried in a vacuum oven at 60°C for 12 hours. The electrodes were punched into circles with a diameter of 12.95 mm. Lithium metal foil with a thickness of 200 μm was punched into circles with a diameter of 14 mm. 2032-type coin cells were fabricated with the addition of 100 μL of 1M LiPF₆ in EC/DEC (v/v=1/1) in an Ar-filled glove box with oxygen and moisture below 1ppm.

8. Figure 5c. Capacity of dry is only 113 mAh g⁻¹, in Figure 5b it is 147 mAh g⁻¹, how come?

Response: The experimental data in Figure 5b is the average of several cells, and the data in Figure 5c is the cycling data of one of them. Figure 5b with error bars (standard error) is depicted below (Figure R1). As one can see, although where rate capability is tested (cycle numbers 0-40) small deviations, the capacities at cycle numbers 41-45 show a rather large deviation. The large difference in the capacities of Figures 5b and 5c is attributed to the degradation of cells and statistical errors.

Figure R1. Figure 5b with error bars.

In response to the reviewer's comment on this matter, we fabricated another set of cells to exclude the effects of degradation. While the results show consistent trends as the data that was previously in the manuscript, the magnitude of the capacity has overall increased due to the elimination of degradation. The increased capacity better fits the results in Figure 5b. The same cyclability test was performed for a new set of cells at 25°C, which also shows decent consistency with Figure 5a. The cycling data at 25°C has been added to Figure 5, and the cycling data at 60°C has been replaced with the new data.

We have also added the following explanation.

[Page 15]

... resistance and coverage. When the cyclability of each cell was evaluated at 25°C and 60°C, both at 0.5C, the different electrode configurations showed similar levels of capacity retention after 100 cycles, with the dry-processed electrodes still exhibiting significantly high discharge capacity, especially at 25°C. (Figure 5c and 5d). ...

Figure 5. Rate capability of pellet, wet, and dry electrodes at (a) 25°C and (b) 60°C. **Cycling test results at (c) 25°C and (d) 60°C.** The mass loading and areal capacity were 23.5 mg cm⁻² and 4.23 mAh cm⁻², respectively.

9. Figure 5. There are many examples in literature where the pellet achieves higher specific capacity than shown here, how come?

Response: The main difference between our pellets and pellets in other literature is thought to be the level of active material loading. Many studies that utilize the pellet type electrodes are focused on the material scale, or particle scale; hence, they are tested in conditions where heterogeneity in the electrodes and kinetics of the Li ions can be neglected. On the other hand, since wet and dry type electrodes are being considered for commercial applications, we compared electrodes at a high mass loading.

To test this, we've fabricated cells with pellet electrodes with a lower mass loading of 6.08 mg cm⁻², compared to the original mass loading of 23.5 mg cm⁻². The cells were tested at both 25°C and 60°C as shown in the following figure (Figure R2). The improved initial capacity and rate capability at high C-rates are noticeable. Also, many papers that have reported higher capacities compared to our pellet electrodes have utilized electrodes with comparably smaller mass loadings, for the reasons described above. The following papers are examples of such cases.

Figure R2. Rate capability of pellet electrodes with low mass loading at 25°C and 60°C.

1. Shin H-J, *et al.* New Consideration of Degradation Accelerating of All-Solid-State Batteries under a Low-Pressure Condition. *Advanced Energy Materials* 13, 2301220 (2023).
2. Kim JT, *et al.* An argyrodite sulfide coated NCM cathode for improved interfacial contact in normal-pressure operational all-solid-state batteries. *Journal of Materials Chemistry A* 11, 20549-20558 (2023)
3. Deng S, *et al.* Insight into cathode surface to boost the performance of solid-state batteries. *Energy Storage Materials* 35, 661-668 (2021).
4. Yi J, He P, Liu H, Ni H, Bai Z, Fan L-Z. Manipulating interfacial stability of LiNi_{0.5}Co_{0.3}Mn_{0.2}O₂ cathode with sulfide electrolyte by nanosized LLTO coating to achieve high-performance all-solid-state lithium batterie. *Journal of Energy Chemistry* 52, 202-209 (2021).

10. Table S1 states i_{1C} as current density but the value given is 5.44 mAh, which has the wrong unit.

Response: The naming of the parameter i_{1C} was written incorrectly. We thank the reviewer for pointing out our mistake. The table will be revised as follows.

i_{1C}	5.44 [mA]	Applied 1C current	calculated
----------	-----------	--------------------	------------

11. Table S1. Why is thickness of cathode 25 μm and electrolyte 140 μm ? In a real cell the cathode would be thicker than the electrolyte. This shows that the model is not representative of a real cell, as mentioned above.

Response: We believe that this can be a question that can be brought up by a lot of readers, and we appreciate the reviewer for bringing the topic up. Before going into the details of SPMs, we want to point out, like in Q6, that the purpose of the model was to neglect different effects and highlight the effects of coverage and solid-state diffusion within the active material particles.

The simplification of an electrode to a single particle is based on the theory of the single-particle model (SPM).^{1,2} This method was originally developed for the application to Pseudo-2-Dimensional (P2D) models, where all effects of gradients in the thickness direction within the electrode are ignored. While the P2D model, by nature, does not require the shortening of the electrode domain, we believe that during the transition from the P2D to 2D model, this step is required and does not interfere with any physics aside from ignoring the gradient in the lateral direction. The figures below are schematics of the P2D model and SPM.

Figure R3. Schematics of P2D model and SPM.

However, we fully understand the concerns about simplifying the model to such an extent. To validate the reliability of the model, the already-elucidated multi-particle model was designed. The model considers ionic conduction in the thickness direction as well as the effects of coverage and solid-state diffusion.

[page 10]

... or ion transport. The thickness of the cathode is decreased, allowing the particle to barely fit into the bulk electrode domain. Such simplifications, although they hamper the ability of the model to precisely pursue real-life cell performance, highlight the effects of coverage while eliminating other effects such as ionic conduction in the bulk electrode domain. In an effort to

...

1. Li J, Adewuyi K, Lotfi N, Landers RG, Park J. A single particle model with

chemical/mechanical degradation physics for lithium ion battery State of Health (SOH) estimation. *Applied Energy* 212, 1178-1190 (2018).

2. Tran NT, Vilathgamuwa M, Farrell T, Choi SS. Matlab simulation of lithium ion cell using electrochemical single particle model. In: 2016 IEEE 2nd Annual Southern Power Electronics Conference (SPEC) (2016).

Response to Reviewer 2

This manuscript reports the effects of shear force in the dry electrode process was systematically investigated by comparing binder-free hand-mixed pellets, wet-processed electrodes, and dry-processed electrodes. Through digitally processed images, the authors quantified a critical factor, ‘coverage’, the percentage of electrolyte-covered surface area of the active materials. The coverage of dry electrodes was significantly higher (67.2%) than those of pellets (30.6%) and wet electrodes (33.3%), enabling superior rate capability and cyclability. Moreover, a physics-based electrochemical model highlighted the effects of solid diffusion by elucidating the impact of coverage on active material utilization under various C-rates. This paper gains insight into the relation between the electrode fabrication process and battery performance, particularly with a focus on the coverage of active materials. This work is possible to be published in Nature Communications after revisions as follows.

Response: We first appreciate the reviewer’s careful review of our manuscript and insightful comments.

1. The authors should give an explanation why the enhanced coverage of dry-processed electrodes could facilitate lithium diffusion and high utilization (capacity) of active materials at high current densities.

Response: We appreciate the comment and have added the following explanation to the manuscript.

[page 12]

... and 91%, respectively. Hence, the difference in lithium diffusion and utilization of active material at high current densities is due to the difference in coverage. Electrodes with low coverage (pellet and wet) have restricted reaction sites and are forced to utilize the non-reacting parts of the active material by diffusion. In extreme cases like 10% coverage, the lithium must diffuse as much as the diameter of the active material to be fully utilized. Dry-processed electrodes with high coverage, on the other hand, have a much shorter diffusion range. When current densities are low the diffusion is not as limiting, while at high current densities, the slow diffusion becomes the limiting factor, therefore lowering the discharge and charge capacities. ...

2. Why the cells were tested at an elevated temperature of 60°C, rather than at room temperature?

Response: We tested the cells at an elevated temperature of 60°C to demonstrate that the coverage and solid-state diffusion can be the kinetically rate determining factors. Considering that sulfide electrolytes have low ionic conductivity at low temperatures, we thought that the sluggish ion conduction could be a rate-determining step competitor, which in turn could interfere with the effects of coverage. However, this does not mean that the effects of coverage and solid diffusion are not visible at room temperature. A new set of cells was fabricated to

demonstrate the capacity and cycling performance at room temperature. The new data will be added to Figure 5.

Also, the following text has been added to the manuscript.

[page 8]

... of 60°C to mitigate the negative effects of the electrolyte's low ionic conductivity at low temperatures, and to utilize ...

Figure 5. Rate capability of pellet, wet, and dry electrodes at (a) 25°C and (b) 60°C. Cycling test results at (c) 25°C and (d) 60°C. The mass loading and areal capacity were 23.5 mg cm⁻² and 4.23 mAh cm⁻², respectively.

3. In Figure 3a, to well separate the corresponding resistance, the EIS spectra should be fitted.

Response: To deconvolute the different resistances mashed up in the EIS data, we've conducted an additional DRT analysis using the previous EIS data. As shown in the following figures, the large impedance at the SE-CAM interface is very noticeable. Such impedance, as already mentioned in the manuscript, is thought to be attributed to the binders coated on the active material surfaces in the wet-processed electrodes.

The results of the analysis will be added to the manuscript as follows.

[page 8]

...in all configurations. The distribution of relaxation time (DRT) method was conducted to deconvolute the various impedances mixed up in the EIS data, as shown in Figure S9. The DRT data can be divided into four different frequency regions:^[41, 42] Warburg diffusion ($10^{-3}\sim 10^{-1}$ Hz), solid electrolyte – anode interface ($10^{-1}\sim 10^1$ Hz), solid electrolyte – cathode interface ($10^1\sim 10^4$ Hz), and solid electrolyte grain boundaries ($10^4\sim 10^6$ Hz). Similar to the EIS data, the DRT analysis also points to the large electrolyte – cathode interface peak in wet-processed electrodes (Figures S9b and S9e), indicating that the large semi-circle in wet electrodes is attributed to the electrolyte – cathode interface. ...

41. Shin SS, *et al.* Quantitative determination of lithium depletion during rapid cycling in sulfide-based all-solid-state batteries. *Chemical Communications* **57**, 3453-3456 (2021).

42. Ye Q, *et al.* Slurry-Coated LiNi_{0.8}Co_{0.1}Mn_{0.1}O₂-Li₃InCl₆ Composite Cathode with Enhanced Interfacial Stability for Sulfide-Based All-Solid-State Batteries. *ACS Applied Materials & Interfaces* **15**, 18878-18888 (2023).

Figure S9. Distribution of relaxation time (DRT) profiles of (a, d) pellet, (b, e) wet-processed, and (c, f) dry-processed electrodes at 25°C and 60°C.

4. To understand solid-state diffusion taking place within active material particles, the authors developed a physics-based electrochemical model, and a thin layer of electrolyte surrounding the particle was introduced. How about the accuracy of this model compared to the

experimental results?

Response: Before we elaborate on the accuracy of the model, we want to briefly emphasize that the purpose of the model is not to accurately model the different electrode fabrication methods, but to elucidate the effects of coverage on the kinetics and the performances of electrodes in ASSBs. As the reviewer may see in the following Figure R4, the model does not accurately pursue the voltage curves of different electrodes at different C-rates. We believe that fabricating such a model requires more information, such as active material percentage, tortuosity, porosity, etc. This model only accounts for the difference in levels of coverage around the active material particles, the diffusion within them, and the ionic conduction in the bulk electrode. We also expanded the SPM to MPM in order to evaluate the effects of ionic conduction in the electrode domain. (Refer to the Q6 of the reviewer 1.)

Figure R4. Voltage profiles of different electrodes and MPM simulations with different coverages.

5. The following related literatures on solid-state batteries are suggested to be cited properly, e.g., *Electrochemical Energy Reviews*, 2023, 6, 17. *Nano Energy*, 2017, 33, 363-386. *ACS Appl. Mater. Interfaces*, 2020, 12, 28345-28350.

Response: We thank the reviewer for the suggestions. The papers mentioned have been added to the manuscript.

14. Tuo K, Sun C, Liu S. Recent Progress in and Perspectives on Emerging Halide

Superionic Conductors for All-Solid-State Batteries. *Electrochemical Energy Reviews* **6**, 17 (2023).

25. Sun C, Liu J, Gong Y, Wilkinson DP, Zhang J. Recent advances in all-solid-state rechargeable lithium batteries. *Nano Energy* **33**, 363-386 (2017).
26. Qiu G, Lu L, Lu Y, Sun C. Effects of Pulse Charging by Triboelectric Nanogenerators on the Performance of Solid-State Lithium Metal Batteries. *ACS Applied Materials & Interfaces* **12**, 28345-28350 (2020).

REVIEWERS' COMMENTS

Reviewer #1 (Remarks to the Author):

Significant changes have been made to the manuscript and my questions have been addressed. It is much improved.

Reviewer #2 (Remarks to the Author):

The authors have addressed the issues proposed by the reviewer so it can be accepted for publication.